# Heterogeneous Manifestations of Epithelial–Mesenchymal Plasticity of Circulating Tumor Cells in Breast Cancer Patients

**DOI:** 10.3390/ijms22052504

**Published:** 2021-03-02

**Authors:** Liubov A. Tashireva, Olga E. Savelieva, Evgeniya S. Grigoryeva, Yuri V. Nikitin, Evgeny V. Denisov, Sergey V. Vtorushin, Marina V. Zavyalova, Nadezhda V. Cherdyntseva, Vladimir M. Perelmuter

**Affiliations:** 1Cancer Research Institute, Tomsk National Research Medical Center, 634050 Tomsk, Russia; tashireva@oncology.tomsk.ru (L.A.T.); grigoryeva.es@gmail.com (E.S.G.); d_evgeniy@oncology.tomsk.ru (E.V.D.); wtorushin@rambler.ru (S.V.V.); zavyalovamv@mail.ru (M.V.Z.); nvch@tnimc.ru (N.V.C.); pvm@ngs.ru (V.M.P.); 2Department of Clinical Biochemistry and Laboratory Diagnostics, Military Medical Academy N.a. S.M. Kirov, 190000 St. Petersburg, Russia; dr.iuriinikitin@gmail.com

**Keywords:** circulating tumor cells, EMT, plasticity, heterogeneity, breast cancer

## Abstract

To date, there is indisputable evidence of significant CTC heterogeneity in carcinomas, in particular breast cancer. The heterogeneity of CTCs is manifested in the key characteristics of tumor cells related to metastatic progression – stemness and epithelial–mesenchymal (EMT) plasticity. It is still not clear what markers can characterize the phenomenon of EMT plasticity in the range from epithelial to mesenchymal phenotypes. In this article we examine the manifestations of EMT plasticity in the CTCs in breast cancer. The prospective study included 39 patients with invasive carcinoma of no special type. CTC phenotypes were determined by flow cytometry before any type of treatment. EMT features of CTC were assessed using antibodies against CD45, CD326 (EpCam), CD325 (N-cadherin), CK7, Snail, and Vimentin. Circulating tumor cells in breast cancer are characterized by pronounced heterogeneity of EMT manifestations. The results of the study indicate that the majority of heterogeneous CTC phenotypes (22 out of 24 detectable) exhibit epithelial–mesenchymal plasticity. The variability of EMT manifestations does not prevent intravasation. Co-expression of EpCAM and CK7, regardless of the variant of co-expression of Snail, N-cadherin, and Vimentin, are associated with a low number of CTCs. Intrapersonal heterogeneity is manifested by the detection of several CTC phenotypes in each patient. Interpersonal heterogeneity is manifested by various combinations of CTC phenotypes in patients (from 1 to 17 phenotypes).

## 1. Introduction

The presence of tumor cells in the peripheral blood largely determines the risk of the development of distant metastases. Detection of circulating tumor cells (CTCs) is a prognostic factor in patients with solid tumors, including different types of breast cancer [1,2,3,4].

Phenotypic heterogeneity of CTCs is already a well-known phenomenon. Obviously, among the many CTC phenotypes, only a few can possess the properties of “seeds”, namely, resistance to anoikis, the ability to extravasate, to transform into disseminated cells, high adaptability to the premetastatic niche, and the ability to induce angiogenesis and stromal formation in metastasis. These cells should be considered as the primary targets of chemotherapy. In connection with the obviousness of such an approach to the prevention of metastases, the task of determining the phenotypic portrait of “seeds” is urgent. One of the most important characteristics of CTCs, ensuring their ability to become metastatic seeds, is the phenotypic heterogeneity of epithelial–mesenchymal (EMT) manifestations [5]. During EMT, a wide range of changes occur, leading to the emergence of many intermediate phenotypes between epithelial and mesenchymal states. It is believed that a significant proportion of CTCs have a hybrid epithelial–mesenchymal phenotype [6].

There are some attempts to divide the hybrid epithelial–mesenchymal pattern of EMT into discrete states. Some authors suggest the division into three [7], four [8] or even five phenotypes [6]. Such variation in the hybrid EMT phenotypes is not accidental and associated with their high plasticity. To date, a consensus has been reached in the interpretation of the essence, genotypic and molecular manifestations of EMT. However, these recommendations also lack clear criteria for assigning tumor cells to discrete EMT phenotypes [9]. According to the recommendation of an international group of researchers who proposed a consensus for determining the status of EMT, one cannot be limited to the use of any set of molecular markers, it is necessary to investigate changes in cellular functions [9] In fact, these conditions can be met only in experimental observations, whereas in clinical trials it is much more difficult to do. Of course, it is possible to study the functional properties of primary tumor cells and CTCs during cultivation in vitro. However, one must take into account the significant distortions in the functional characteristics of tumor cells during their adaptation to survival in vitro. Thus, there are significant objective difficulties in determining the characteristics of stable phenotypic variants of CTCs. However, given the limited information obtained using any set of EMT markers, the study of heterogeneous CTC phenotypes in breast cancer patients and their association with the most important features of tumor progression has great perspectives.

## 2. Results

### 2.1. Frequency Analysis of CTCs with Co-Expression of Epithelial Markers Depending on Snail, N-Cadherinn, and Vimentin Expression

In our study, the incidence of CTCs was 71.8% (28/39). The frequencies of CTCs with different combinations of EpCAM and CK7 expression do not differ from each other and vary from 53.2% to 76.6%.

The relationship between the expression of epithelial markers (such as EpCAM and CK7) and molecules reflecting the initiation of EMT and occurrence of hybrid/mesenchymal EMT phenotypes (Snail, N-cadherin, Vimentin) was complex. Thus, in breast cancer patients, the cases with the CD45–EpCAM+CK7+N-cadherin– CTC phenotype is significantly less than cases with the CD45–EpCAM+CK7–N-cadherin– CTCs. Such differences were observed regardless of the Vimentin expression in these CTCs (Figure 1). A different pattern was observed when comparing the frequencies of CK7 expression in CTCs with positive expression of N-cadherin and EpCAM. The frequency of cases with CTCs with or without CK7 expression did not differ. The frequency of CK7-positive CTCs with positive expression of N-cadherin was associated with Vimentin expression. In the absence of Vimentin expression, the frequency of CD45–EpCAM+CK7+N-cadherin+ phenotype CTCs was lower than CD45–EpCAM+CK7–N-cadherin+. In case of positive Vimentin expression in CD45–EpCAM+CK7+N-cadherin+ CTCs, there were no differences. The described phenomenon of the relationship between the frequencies of CK7+ and CK7– EpCAM-positive CTCs on the presence of N-cadherin expression did not depend on Snail expression. Comparison of the frequencies of CD45–EpCAM–CK7+ and CD45–EpCAM+CK7– CTCs allows us to conclude that the frequency of detecting such populations is not associated with the expression of Snail and N-cadherin. Taking into account the expression of Vimentin in the compared populations of CTCs, this conclusion remains the same. The exception is the CD45–EpCAM–CK7+Snail+N-cadherin+Vimentin– CTCs, which is less common than the CD45–EpCAM+CK7–Snail+N-cadherin+Vimentin– population.

### 2.2. Quantitative Analysis of CTC Subsets with Heterogeneous EMT Manifestations

Co-expression of the EpCAM, CK7, Snail, N-cadherin and Vimentin was analyzed to establish the association of the CTC phenotypes with their number (Table 1).

Relatively large number of CTCs had the following phenotypes: CD45–EpCAM+CK7–Snail–N-cadherin–Vimentin– (7.05 (0.83–17.01) cells per ml), CD45–EpCAM–CK7+Snail–N-cadherin–Vimentin+ (3.73 (0.83–7.07) cells per ml), CD45–EpCAM–CK7+Snail–N-cadherin–Vimentin– (1.66 (0.00–5.39) cells per ml), CD45–EpCAM+CK7–Snail–N-cadherin–Vimentin+ (0.83 (0.00–1.66) cells per ml). In addition, sufficiently high level of CTCs with negative expression of Snail or N-cadherin and lack of EpCAM was observed (CD45–EpCAM–CK7+Snail–N-cadherin+Vimentin+ – 1.24 (0.00–4.56) cells per ml, CD45–EpCAM–CK7+Snail+N-cadherin–Vimentin+ – 0.41 (0.00–1.66) cells per mL). We combined these phenotypes into group 1. The number of the remaining 18 CTC phenotypes was minimal with a median of 0.00 cells per ml. These phenotypes were included in group 2 (Figure 2). Regardless of the Snail, N-cadherin and Vimentin expression EpCAM+CK7+ CTCs were found in the smallest quantities (Table 1, column 2).

All patients had CTC with phenotypes from group 1, while the incidence of CTC phenotypes belonging only to group 2 was 7.14% (2/28). In 92.86% of cases, a combination of CTCs belonging to group 1 and 2 was observed. Comparison of the CTC number in two groups, showed following significant differences. The largest number of CTCs was observed in the first group (1.66 (0.00–5.60) cells per mL) and the smallest—in the second (0.00 (0.00–0.00) cells per 1 mL) (*p* < 0.0001) (Figure 2B). According to the heat map, CTCs reaching high values in individual patients are highly variable in number.

### 2.3. Interpersonal Heterogeneity of the CTC Subsets

The interpersonal heterogeneity of CTC characteristics was most clearly illustrated by the fact that among the 28 studied cases, there were no two cases with the same set of CTC phenotypes. Heterogeneity was also confirmed by cluster analysis of the number of CTCs with different phenotypes, which allowed us to divide all cases into two clusters (Figure 3).

The clusters differed in the total number of CTCs and was 73.45 (48.14–84.86) cells per ml in the first cluster versus 8.88 (6.35–17.97) cells per ml in the second cluster, *p* = 0.000). The number of CTC phenotypes (in the first cluster 11 (8–14) versus 5.5 (3–6) in the second cluster, *p* = 0.000).

The second cluster differed from the first not only in the smaller number of CTCs phenotypes, but also in the qualitative composition (Figure 4). Among the CTC phenotypes found in patients assigned to the second cluster, CTCs with the following phenotypes: CD45–EpCAM+CK7–Snail–N-cadherin–Vimentin– (*p* = 0.015), CD45–EpCAM–CK7+Snail+N-cadherin+Vimentin+ (*p* = 0.004), CD45–EpCAM+CK7–Snail–N-cadherin+Vimentin– (*p* = 0.046), and CD45–EpCAM+CK7+Snail+N-cadherin+Vimentin+ (*p* = 0.001) were less common. Despite the absence of significant differences, it is worth noting that following CTCs were absent among the phenotypes of the second cluster: CD45–EpCAM+CK7+Snail+N-cadherin+Vimentin–, CD45–EpCAM+CK7+Snail–N-cadherin–Vimentin+, CD45–EpCAM+CK7+Snail–N-cadherin+Vimentin–, CD45–EpCAM–CK7+Snail+N-cadherin+Vimentin–, CD45–EpCAM+CK7+Snail+N-cadherin–Vimentin+, four of which were characterized by co expression of EpCAM and CK7.

Clinical parameters of patients (age, menopausal status, stage, grade, lymph node metastasis) were not associated with the patient’s belonging to one of the two cluster by variants of the combination of CTC phenotypes.

Comparison of two clusters of patients by the representation of small and numerous groups of CTCs showed, that CTCs belonging to two groups were found more often in patients of both clusters (100% and 85.71%). Cases with CTC phenotypes from the first group (14.29%) belonged to the second cluster. Not a single case had been found when the CTC phenotypes belonged only to the second group (Figure 5).

Figure 6 illustrates the interpersonal heterogeneity in the number of CTC phenotypes in each of the patients belonging to the cluster 1 and cluster 2.

Interpersonal heterogeneity was more pronounced in cases belonging to the first cluster. Wherein in the first cluster cases with six or less different CTC phenotypes in one patient did not occur, in the second cluster they prevailed (0/8 and 6/9, respectively, *p* = 0.009).

### 2.4. The Phenotypes of Tumor Cells in the Primary Tumor and in the Peripheral Blood

Comparison of the frequency and number of tumor cells in the primary tumor and among CTCs was carried out by detecting of the co-expression of CK7, EpCAM, and N-cadherin in the same cell (Figure 7). The using of only three markers is due to the limitations of the method, which do not allow the simultaneous detection of a larger number of molecules in the tissue slides.

In total, six phenotypes of cells were studied with different combinations of these markers. It turned out that tumor cells of all six phenotypes were found both in the tumor and in the blood. The frequencies of different cell phenotypes in the tumor were in the range from 30.3% to 87.8%, and from 17.9% to 69.2% in the peripheral blood. Comparison of the relative number of primary tumor cells and CTCs of identical phenotypes indicates the presence of two alternative variants. The proportion of the EpCAM–CK7+N-cadherin– tumor cells was significantly higher among CTCs compared to the primary tumor; on the contrary, the percentage of EpCAM+CK7+N-cadherin+ cells was higher in the primary tumor.

### 2.5. Association of CTC Phenotypes with Clinicopathological Characteristics of Breast Cancer

As mentioned above, attempts to establish an association between clinical and morphological characteristics with any variant of grouping of CTC phenotypes in patients have led to the conclusion that there is no such relationship. Despite this, associations of some clinicopathological parameters with the frequency and number of CTC phenotypes were found.

Among of the 24 CTC phenotypes, the number of CTCs belonging to 21 phenotypes was not associated with the size of the primary tumor. With increasing of the size of primary tumor, the number of CTCs in the following three populations increased: CD45–EpCAM+CK7–Snail–N-cadherin–Vimentin–, CD45–EpCAM+CK7–Snail–N-cadherin–Vimentin+, and CD45–EpCAM–CK7+Snail–N-cadherin–Vimentin– (Table 2). With increasing of the tumor grade from T1 to T2, among the 24 discussed CTC populations, the number of CTCs with CD45–EpCAM+CK7+Snail–N-cadherin+Vimentin+ phenotype decreased (0.00–0.83) cell per ml and 0.83 (0.83–2.22) cells per ml, respectively, *p* = 0.023). The number of CD45–EpCAM+CK7+Snail–N-cadherin+Vimentin+ CTCs was found to be associated with the cell proliferative activity in the primary tumor. It turned out that when the level of proliferative activity in the primary tumor, assessed by Ki-67 expression was higher than 20%, the absolute number of CTCs with the CD45–EpCAM+CK7+Snail–N-cadherin+Vimentin+ phenotype was significantly lower in comparison with cases when the proliferative activity was below 20% (0.83 (0.00–1.66) cell per ml and 0.00 (0.00–0.00) cell per ml, respectively, *p* = 0.022).

Comparison of the frequencies (Table 3) and number (Table 4) of 24 studied CTC phenotypes showed certain differences between the groups of patients with and without lymphogenous metastasis. In patients with lymphogenous metastases, three CTC populations were more common, namely CD45–EpCAM–CK7+Snail+N-cadherin+Vimentin+, CD45–EpCAM+CK7+Snail+N-cadherin+Vimentin– and CD45–EpCAM+CK7–Snail+N-cadherin+Vimentin+ (*p* = 0.03). The phenotypic diversity of CTCs in the groups with and without lymphogenous metastasis did not differ and varied from 2 to 17 phenotypes in one patient in the absence of metastases and from 1 to 15 different phenotypes in one patient in the presence of lymphogenous metastasis.

## 3. Discussion

It is believed that the EMT makes the tumor cell more invasive and promotes intravasation and the formation of circulating tumor cells (CTCs), which ultimately leads to the development of metastases [10]. CTCs in patients with breast cancer are considered to be a strong prognostic factor, since they could be used as a marker of hematogenous spread of tumor cells [11]. Moreover, epithelial-type CTCs had the strongest metastases formation ability, comparing with mesenchymal-type CTCs [5]. Circulating tumor cells in breast cancer can exhibit hybrid epithelial/mesenchymal states. According to the study of Yu, M. et al. (2013), this phenomenon reflects the heterogeneity of the cellular composition of the primary tumor [6]. For a long period, various manifestations of EMT-MET were considered as sequential molecular genetic events. This approach was supported by the obtained facts about certain time periods of the expression of various genes and proteins, reflecting the development of EMT-MET. The earliest event of EMT is the appearance of SNAIL expression, which is shown in both, normal and tumor tissues. Thus, treatment of the normal epithelium of the mammary gland with TGFβ, which act as EMT inductor, at the first stage, ZEB1 and SNAIL expression was observed, the switch from the E-cadherin expression to N-cadherin was detected later [12]. In various carcinomas, the role of SNAIL is to induce EMT in primary tumors [13]. RY-J Huang et al. (2013) proposed four subgroups of EMT manifestations and believe that SNAIL is expressed during EMT at first moment, while N-cadherin, in the absence of E-cadherin, is regarded as a mesenchymal trait [8].

To characterize the EMT phenotype of breast cancer cell line HMLER, the researchers consider the expression of the CDH1 and EPCAM genes as epithelial, hence, early traits, and VIM and ZEB2 as mesenchymal and, therefore more later traits [8]. While EMT, membrane expression of EpCAM is lost and protein translocated into the nucleus [14]. In many studies, the inhibition of EpCAM expression in EMT is noted [15]. However, it is not always clarified whether the loss of membrane EpCAM expression and its intracellular translocation occurs, or the cell completely loses EpCAM, as was observed in our study. The place of these changes in EMT dynamics is indirectly indicated by the data that EpCAM is involved in the regulation of EMT by suppressing the SNAIL expression [16,17]. Ye X, et al. (2015) also noted a negative correlation between EpCAM and Snail expression [17]. Since Snail expression is observed immediately after induction of EMT, suppression of EpCAM, apparently, should occur thereafter, also in the initial phase of EMT.

Regarding the expression of cytokeratins and vimentin, it is argued that maintaining certain levels of cytokeratin expression indicates that these cells have not undergone EMT. It is believed that, as a rule, the transition from the expression of cytokeratins to vimentin is observed after the completion of EMT [9]. Relatively CK7 Shaolei Lu et al. suggest that the loss of CK7 in breast cancer cells may be associated with EMT and with increase of tumor stemness [18].

Considering the above, in our study, the expression of the Snail with retention of EpCAM and/or CK7 expression can be considered as the initial manifestations of EMT with preservation of epithelial features. Then, Snail expression with loss of EpCAM or CK7 expression is considered as later manifestations of EMT with loss of some epithelial properties and expression of N-cadherin and Vimentin, as a manifestation of mesenchymal traits. Any combination of the retention of epithelial traits and the appearance of mesenchymal ones, apparently, can be considered as manifestations of epithelial–mesenchymal plasticity.

There are numerous attempts to separate different variants of EMT phenotypes into epithelial, hybrid, and mesenchymal. It is proposed to divide into three, four, or five groups of phenotypes [6,7,8]. However, the criteria for separation are not clear enough. This is due to the complexity of the processes taking place during EMT-MET.

In accordance with the conclusions of the consensus group on agreeing on the nomenclature of terms and concepts of EMT mechanisms, this process is not linear, but rather multilevel and multivariate. In this regard, the division of EMT into phases is very difficult and suppositive. The concept of epithelial–mesenchymal plasticity (EMP) is recommended, which is intermediate between epithelial and mesenchymal phenotypes. EMP suggests the possibility of changing phenotypes towards both, epithelial and mesenchymal phenotypes. However, nothing is said about the possibility to determine whether it is a manifestation of EMT or MET when a hybrid epithelial–mesenchymal phenotype is detected [9]. In our study, 28.2% of patients had no CTCs. The CTCs in the remaining patients were characterized by pronounced heterogeneity in EMT characteristics.

Depending on the expression of EpCAM, CK7, Snail, N-cadherin, and Vimentin, we considered the probability of 24 CTC phenotypes in breast cancer patients. Only CD45–EpCAM+CK7+Snail+N-cadherin–Vimentin– phenotype of CTCs were not found in the blood of breast cancer patients. The remaining 23 variants of CTC phenotypes were identified in the patients, and this manifested intrapersonal CTC heterogeneity. Only CD45–EpCAM+CK7+Snail–N-cadherin–Vimentin– CTCs, which are found in 15.4% of patients, probably do not have EMT manifestations or have initial EMT manifestations that are not detected by the set of markers used by us. However, at the same time tumor cells of the primary tumor with such a phenotype had the ability to intravasate, apparently unrelated to EMT and the invasive mechanism of intravasation [18].

The 22 different CTC phenotypes found in patients with breast cancer are probably different variants of hybrid (epithelial/mesenchymal) states that demonstrate different manifestations of EMP. In this case, apparently, among the different mechanisms of intravasation in the primary tumor [19] and the appearance of CTCs, the mechanisms associated with EMT and the ability of tumor cells to mesenchymal mechanism of invasion are prevail. It is obvious that the variability of EMP manifestations in 22 CTC subpopulations does not affect their essential properties, which limit the ability of tumor cells to intravasate. At the same time, the number of CTCs varied significantly.

It is well known that the number of CTCs largely determines the probability of developing metastases. There is strong evidence that only 0.01% of all CTCs are capable to establish clinically significant metastases. In this regard, one of the important issues in the CTCs studies is understanding of the mechanisms that determine their number. According to the literature and the results of our studies, it is well known that the CTC population is extremely heterogeneous. Therefore, it is not so much the total number of CTCs that is important as the number of CTC subsets that can act as “seeds” in the development of metastases.

The quantitative evaluation of CTCs made it possible to divide all the detected phenotypes into numerous and rare. Probably, numerous CTC subsets have a more pronounced potential for intravasation and/or less potential for extravasation than CTCs found in minimal amounts. This may also be due to the better survival of numerous CTC subsets and the death of small CTC subpopulations. It should be noted that the number of eight EpCAM+CK7+ CTC subpopulations in all patients is extremely small, regardless of the presence or absence of other markers. Since CTCs remain in the blood for an extremely short time, it is most likely that their phenotype is extremely close to the phenotype corresponding to the population of primary tumor cells, which, after intravasation, became CTCs. Thus, a comparison of the CTC phenotypes with their number while simultaneously studying the primary tumor makes it possible to reveal the presence of one of the key factors that determine the likelihood and number of CTCs of the similar phenotype. It could be expected that a certain spectrum of EMT manifestations may be associated with the ability of primary tumor cells to intravasate. In our study, out of six markers which characterized CTCs, five reflect manifestations of EMT. Positive expression of Snail, N-cadherin, Vimentin and loss of EpCAM and/or CK7 expression should indicate the development of EMT. Moreover, different variants of co-expression of these markers can correspond to different stages of this process. In our study, expression of three markers (CK7, EpCAM, N-cadherin) was evaluated in the primary tumor. Frequency of EpCAM+CK7+ tumor cells in the primary tumor and in the blood did not differ statistically. Thus, the small number of eight subpopulations of EpCAM+CK7+ CTCs may be due not so much to a reduced capacity for intravasation, but to a small number of such cells in the primary tumor. The population of CTC with the CD45–EpCAM+CK7–N-cadherin– phenotype is of great interest. This population was the most numerous and occurred more often both in the tumor and in the peripheral blood. It should be noted that, additional assessment of the EMT markers Snail and vimentin, revealed that the CD45–EpCAM+CK7–Snail–N-cadherin–Vimentin– CTC subpopulation was also still the most numerous. Apparently, this population of CTCs corresponds rather to the epithelial phase of EMT, since the expression of Snail, N-cadherin, and Vimentin is absent. It is necessary to clarify what place the lack of CK7 expression in CTCs occupies in the development of EMT and in successful intravasation, as well as resistance to cell death in peripheral blood.

Comparison of the relative number of cells in the primary tumor and among the CTCs suggests the potentials of cells with different phenotypes for intravasation and survival in the blood flow. In our study, the expression of three markers (CK7, EpCAM, N-cadherin) was evaluated in the primary tumor. Of great interest are CD45–EpCAM–CK7+N-cadherin– and CD45–EpCAM+CK7+N-cadherin+ CTC subpopulations. Since CTCs with the CD45–EpCAM–CK7+N-cadherin– phenotype were significantly higher among CTCs in comparison with the primary tumor, it can be assumed that they have a pronounced capacity to intravasate and/or survive. Conversely, CD45–EpCAM+CK7+N-cadherin+ cells have a poorer ability to intravasate and/or survive in blood flow. It would be useful to find out which role plays the absence of EpCAM expression in CTCs in the development of EMT and successful intravasation, as well as in resistance to cell death in peripheral blood.

Interpersonal heterogeneity manifested itself in a different number of CTC phenotypes in patients (from 1 to 17 phenotypes). Interpersonal heterogeneity was so pronounced that among the studied patients there were no identical cases with the same combination of CTC phenotypes. Since all of the detected CTC phenotypes are more likely to belong to different manifestations of EMP, the number of CTC phenotypes that are identified in each case may be important. Apparently, the probability that CTCs have the properties of “seeds” is higher in patients with 17 phenotypes than in the case when there is only 1 CTC phenotype. Since the study is prospective, the significance of the number of CTCs, as well as other characteristics of CTCs, for assessing the risk of hematogenous metastasis remains to be clarified.

Among 23 phenotypes of CTCs, the number of CTCs in three populations increased with an increase in the size of the primary tumor: CD45–EpCAM+CK7–Snail–N-cadherin–Vimentin–, CD45–EpCAM+CK7–Snail–N-cadherin–Vimentin+, and CD45–EpCAM–CK7+Snail–N-cadherin–Vimentin–. The lost expression of one of the epithelial markers makes similar these CTC subsets, as well as the absent expression of N-cadherin and Snail. Probably, the increase in the size of the primary tumor entails an increase in the number of cells in these three subsets the number of similar CTCs. If the number of cells in these subsets is not increased in a primary tumor of a larger size, an increase in the number of CTCs is possible due to improved conditions for intravasation. The relationship between the presence of CTCs and tumor size without taking into account their heterogeneity was noted by Janni W et al. (2013) [20].

The number of CTCs with CD45–EpCAM+CK7+Snail–N-cadherin+Vimentin+ phenotype had an inverse connection with grade and proliferative activity of the primary tumor. Probably, a small number of these CTCs can be associated with a small number of cells with this phenotype in the primary tumor due to the more pronounced proliferation of other competitive and high-grade clones of the heterogeneous tumor. On the other hand, an increase in proliferative activity in high-grade tumors may be associated with a worsening of conditions for intravasation of cells with CD45–EpCAM+CK7+Snail–N-cadherin+Vimentin+ phenotype.

Circulating tumor cells probably cannot be considered as “seeds” for lymphogenous metastases. Otherwise, it should be recognized that metastases in the lymph nodes occur hematogenously. Despite a certain probability of the hematogenous nature of the development of lymph metastases, there is convincing evidence of a difference in the mechanisms of lymphogenous and hematogenous metastasis [21].

Our study showed that frequency of three CTC populations (CD45–EpCAM–CK7+Snail+N-cadherin+Vimentin+, CD45–EpCAM+CK7+Snail+N-cadherin+Vimentin–, CD45–EpCAM+CK7–Snail+N-cadherin+Vimentin+) in patients with lymphogenous metastases was from 3 to 10 times higher than in patients without lymphogenous metastases. Such relationship between the presence of CTCs (without dividing into different phenotypes) with lymphogenous metastasis in breast cancer was noted earlier [1,20].

Moreover, it was found that the presence of CTCs reduced disease-free survival of patients with lymphogenous metastases but was not associated with negative lymph node status [22]. It should be emphasized that despite the small number of cells with co-expression of EpCAM and CK7, cells with the CD45–EpCAM+CK7+Snail–N-cadherin+Vimentin+ phenotype were associated with proliferative activity in the primary tumor, while CD45–EpCAM+CK7+Snail+N-cadherin+Vimentin– cells were associated with lymphogenic metastasis. Therefore, the prognostic value of CTCs with different phenotypes is not always related to their number in peripheral blood.

Apparently, tumor cells of these three subpopulations have the ability to intravasate not only into blood vessels, but also into lymphatic vessels. These three populations of CTCs combine EMT traits, which are manifested by a set of the “early” EMT marker – Snail and the “late” EMT marker – N-cadherin, and in two cases also by the expression of Vimentin. These manifestations of EMT may correspond to a hybrid state, which suggests the possibility of implementing the variant of intravasation by means of the mesenchymal mechanism of invasion [19].

## 4. Materials and Methods

### 4.1. Patients

The prospective study included 39 patients with invasive breast carcinoma of no special type (IC NST) T1-2N0-2M0, admitted for treatment to Cancer Research Institute, Tomsk National Research Medical Center. No-one was treated by neoadjuvant chemotherapy. Venous ethylenediaminetetraacetic acid (EDTA) blood samples were taken one to two days before surgical intervention. The study was approved by the Local Committee for Medical Ethics of our institute (17 June 2016, the approval No. 8), and informed consent was obtained from all patients prior to analysis. The clinicopathological parameters of the patients with breast cancer are presented in Table 5.

### 4.2. CTC Enrichment and Flow Cytometry

Limitations of CellSearch system and microfluidics-based approaches as CTC-enrichment methods have been described previously [23]. As EDTA accelerates the deposition of red blood cells, it is possible to obtain plasma containing many non-erythrocyte cells including CTC. Therefore, to avoid the loss of target cells, we chose this method for producing a cell concentrate by incubating the EDTA blood in a thermostat at 37 °C for 90 min and analyzing the total number of nucleated cells obtained from 12 mL of whole blood. After the cell concentration step and centrifugation, the cells were resuspended in 1 mL of staining buffer (Sony Biotechnology, Tokyo, Japan).

The phenotypic characteristics of CTCs and EMT features was analyzed on the Novocyte 3000 (ACEA Biosciences, San Diego, CA, USA). Cells were stained in two steps: surface markers were first stained, and then intracellular markers were stained. After Fc-blocking of cell concentrate by Human TruStain FcX™ Fc Receptor Blocking Solution (Biolegend, San Diego, CA, USA), monoclonal antibodies were added: 5 μL of BV570-anti-CD45 (clone HI30, mouse IgG1, Sony Biotechnology, San Jose, CA, USA), BV650-anti-EpCAM (clone 9C4, mouse IgG2b, Sony Biotechnology, San Jose, CA, USA) and PE-Cy7-anti-N-cadherin (clone 8C11, mouse IgG1, Sony Biotechnology, San Jose, CA, USA). Viable cells were identified by the use of 7-amino-actinomycin D (7-AAD, Sony Biotechnology, San Jose, CA, USA). After incubation, erythrocytes were lysed by OptiLyse buffer (Beckman Coulter, Marseille, France) and washed in Cell Wash buffer (BD Biosciences, San Jose, CA, USA). For intracellular stains, cells were permeabilized using BD Cytofix/Cytoperm (BD Biosciences, San Jose, CA, USA). After permeabilization and washing, monoclonal antibodies were added: BV650-anti-EpCAM (clone 9C4, mouse IgG2b, Sony Biotechnology, San Jose, CA, USA), AF647-anti-Cytokeratin 7/8 (clone CAM5.2, Mouse IgG2a, BD Pharmingen, San Jose, CA, USA), AF488-anti-Snail (clone 20C8, Mouse IgG2a, eBioscience, Thermo Fisher Scientific, San Jose, CA, USA) and AF750-anti-Vimentin (clone 280618, Rat IgG2A, R&D Systems, Minneapolis, MN, USA).

The antibody quality control was performed. The isotype control antibodies at the same concentration were added to the control sample. MCF-7 cells were used as positive control and U937 cells were used as negative control. CTC spiking experiment was performed. The CTCs were isolated from blood of the breast cancer patient using a cell sorter, MoFlo XDP, with Summit software (Beckman Coulter, Miami, FL, USA) and were spiked into the whole blood of a healthy donor at 18–33 cells in 100 μL of blood (spike-in CTC samples, n = 3).

The gating of the cell populations was carried out on the basis of determining the parameters of forward scatter light signals (FSC) and side scatter light signals (SSC). Cells were then analyzed for fluorescence in density- and dot-plot modes. Positive cells were counted per 1 mL of whole blood.

### 4.3. Multiplex Immunofluorescence (mIF) Staining

Multiplex immunofluorescence was used for the evaluation of epithelial and mesenchymal markers in tumour cell in primary lesions. Multiplex IF was performed by using the Bond RXm staining system (Leica Microsystems, GmbH, Wetzlar, Germany). The following antibody-Opal pair and dilutions were used: Anti-EpCAM-Opal 520 (1:1000, clone E144, Abcam, Cambridge, UK), Anti-N-cadherin-Opal 650 (1:500, clone EPR1791-4, Abcam, Cambridge, UK), and Anti-CK7-Opal 690 (1:10, OV-TL 12/30, Agilent, Santa-Clara, CA, USA). The slides were counterstained with DAPI for 5 min and mounted with VECTASHIELD Hard Set (Vector Labs, Burlingame, CA, USA). All primary antibodies were optimized using control tissues. Slides were scanned using the PerkinElmer Vectra (Vectra 3.0.3; PerkinElmer, Waltham, MA, USA). Tissue imaging and analysis were performed using inForm Advanced Image Analysis software (inForm 2.2.1; PerkinElmer, Waltham, MA, USA) according to the recommendations [24].

The frequency and count of the following cell populations were scored: EpCAM+CK7–N-cadherin–; EpCAM+CK7–N-cadherin+; and EpCAM+CK7+N-cadherin–; EpCAM+CK7+N-cadherin+; and EpCAM–CK7+N-cadherin–; EpCAM–CK7+N-cadherin+.

### 4.4. Statistical Analysis

The data were analyzed with the statistical software STATISTICA 13 (StatSoft, UK, USA) and GraphPad Prism 8.3.1 (GraphPad Software, San Diego, CA, USA). The Shapiro–Wilk test was used. The U-criterion Mann–Whitney was used to compare the differences in the number of CTC subsets. Spearman’s test was applied to search for correlations between the number of CTCs in different subsets. A comparison of the frequencies of CTCs with different phenotypes, as well as the frequencies of CTC phenotype combinations, was performed using the two-sided Fisher test. Cluster analysis was used to divide patients into groups based on the combination of CTC subsets. *p* < 0.05 was considered statistically significant.

## 5. Conclusions

The results of this work indicate that the CTCs in breast cancer are characterized by pronounced heterogeneity in the manifestations of the EMT. Heterogeneity can be both intra and interpersonal. Taking into account the presence or absence of epithelial and mesenchymal markers used in our study, most of the detected CTC phenotypes can be attributed to hybrid ones. The wide range of different EMT phenotypes of CTC suggests that the degree of EMT is not an obstacle to intravasation. However, we have shown that the number of CTCs belonging to different populations is largely determined by their phenotype and is not proportional to the number of similar populations in the primary tumor, which may be due to the different ability to intravasate and survive in the blood. The association between some CTC phenotypes and tumor size, proliferative activity, and lymphogenic metastasis suggests that, as the duration of prospective follow-up increases, it will be possible to identify CTC phenotypes associated with hematogenous metastasis from the detected ones.

## Figures and Tables

**Figure 1 ijms-22-02504-f001:**
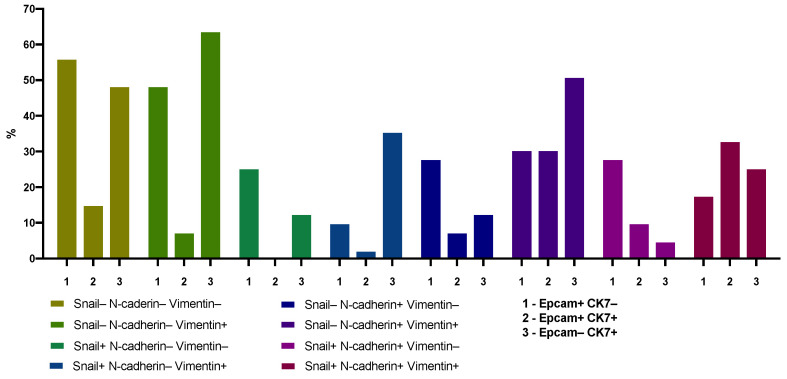
The frequencies of circulating tumor cells (CTCs) with co-expression of EpCAM and CK7 depending on Snail, N-cadherin and Vimentin expression in breast cancer patients.

**Figure 2 ijms-22-02504-f002:**
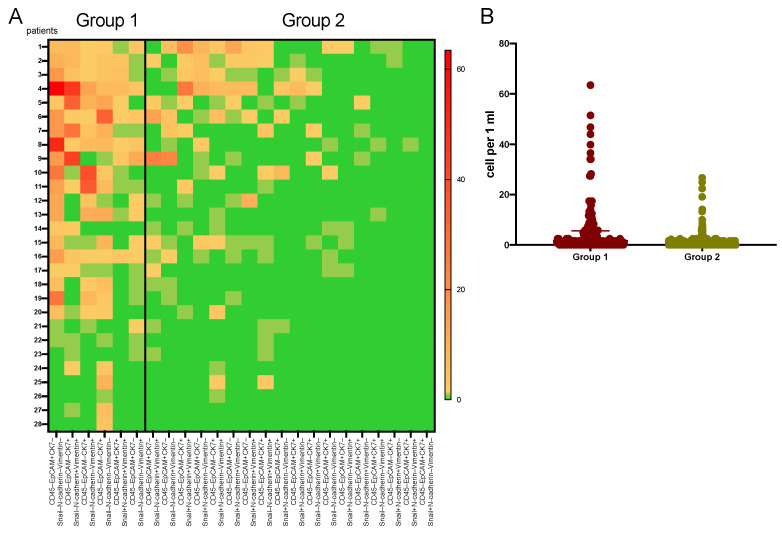
Heat map of the number of CTCs with various phenotypes. CTC phenotypes are located from left to right in decreasing order of their number. The frequency of occurrence (**A**) of each of the CTC phenotypes located from left to right in decreasing order of their maximal number. Vertical line divide CTC phenotypes by two groups (the numerous and the small). The median number of the CTCs from groups (**B**).

**Figure 3 ijms-22-02504-f003:**
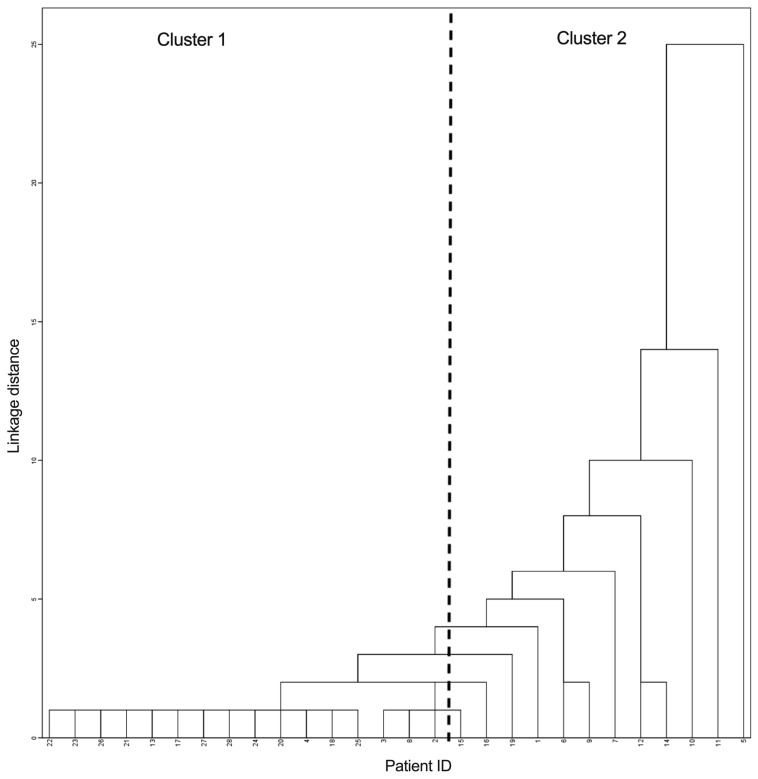
Dendrogram of breast cancer patients by the number of CTCs with different phenotypes.

**Figure 4 ijms-22-02504-f004:**
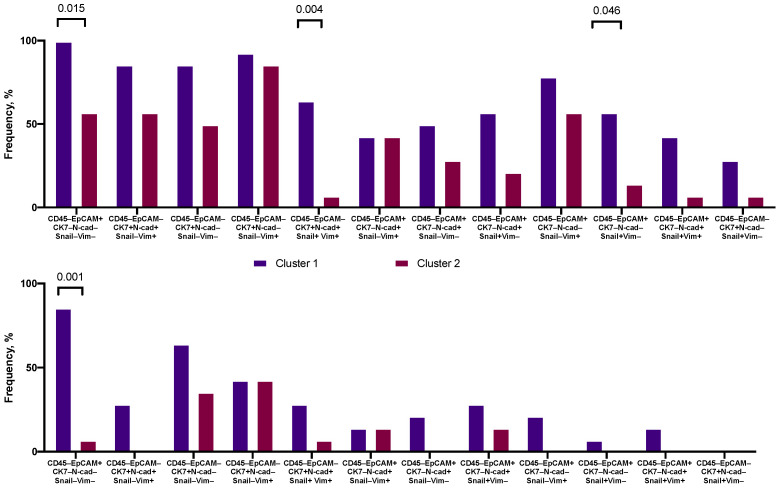
Frequency of occurrence of CTCs with different phenotypes in patients belonging to the first and second clusters. N-cad—N-cadherin, Vim—Vimentin.

**Figure 5 ijms-22-02504-f005:**
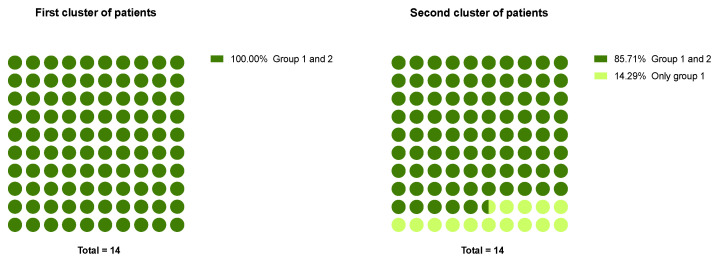
Variants of CTC phenotypic combinations in two clusters of breast cancer patients.

**Figure 6 ijms-22-02504-f006:**
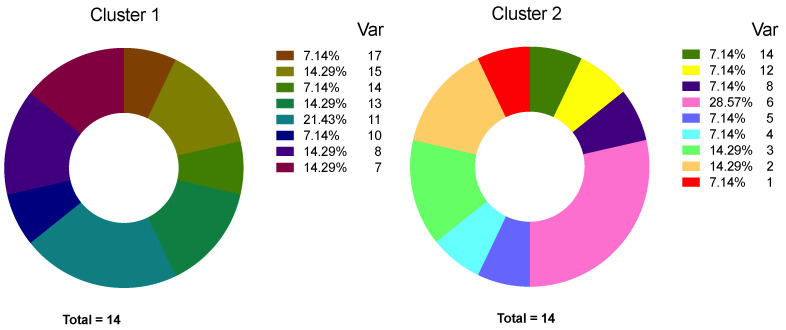
Interpersonal heterogeneity in the number of phenotypes in each of the patients belonging to the cluster 1 and 2. Var—variability of number of CTC phenotypes. N—number of patients.

**Figure 7 ijms-22-02504-f007:**
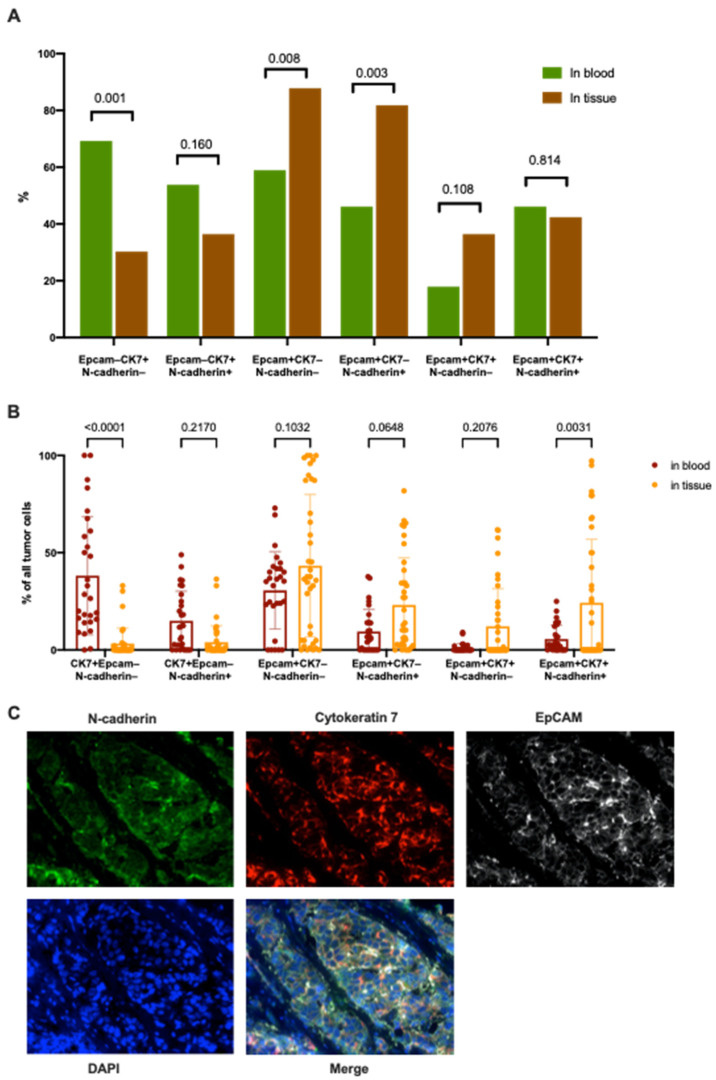
The frequencies (**A**) and number (**B**) of tumor cells in the primary tumor and among CTCs which co-express of CK7, EpCAM, and N-cadherin in the same cell. Invasive breast carcinoma cells stained by CK7, EpCAM, and N-cadherin (**C**).

**Table 1 ijms-22-02504-t001:** The number of different CTC subsets with heterogeneous manifestations of EMT.

		EpCAM+ CK7−	EpCAM+ CK7+	EpCAM− CK7+
		1	2	3
Snail– N-cadherin– Vimentin–	a	7.05 (0.83–17.01)	0.00 (0.00–0.00) p1–2 = 0.000	1.66 (0.00–5.39) p1–3 = 0.007 p2–3 = 0.000
Snail– N-cadherin– Vimentin+	b	0.83 (0.00–1.66) *p* = 0.000	0.00 (0.00–0.00) p1–2 = 0.000 *p* = 0.115	3.73 (0.83–7.07) p1–3 = 0.001 p2–3 = 0.000 *p* = 0.126
Snail+ N-cadherin– Vimentin–	c	0.00 (0.00–0.83)	0.00 (0.00–0.00) p1–2 = 0.005	0.00 (0.00–0.00) p1–3 = 0.091 p2–3 = 0.043
Snail+ N-cadherin– Vimentin+	d	0.00 (0.00–0.00) *p* = 0.012	0.00 (0.00–0.00) p1–2 = 0.117 *p* - no data	0.41 (0.00–1.66) p1–3 = 0.002 p2–3 = 0.001 *p* = 0.021
Snail– N-cadherin+ Vimentin–	e	0.00 (0.00–0.83)	0.00 (0.00–0.00) p1–2 = 0.008	0.00 (0.00–0.00) p1–3 = 0.007 p2–3 = 0.248
Snail– N-cadherin+ Vimentin+	f	0.00 (0.00–0.83) *p* = 0.139	0.00 (0.00–0.83) p1–2 = 0.529 *p* = 0.010	1.24 (0.00–4.56) p1–3 = 0.015 p2–3 = 0.001 *p* = 0.000
Snail+ N-cadherin+ Vimentin–	g	0.00 (0.00–0.83)	0.00 (0.00–0.00) p1–2 = 0.019	0.00 (0.00–0.00) p1–3 = 0.003 p2–3 = 0.201
Snail+ N-cadherin+ Vimentin+	h	0.00 (0.00–0.41) *p* = 0.352	0.00 (0.00–1.66) p1–2 = 0.151 *p* = 0.002	0.00 (0.00–1.24) p1–3 = 0.130 p2–3 = 0.733 *p* = 0.005

**Table 2 ijms-22-02504-t002:** The number of different CTCs subsets with heterogeneous manifestations of epithelial–mesenchymal (EMT).

CTC Phenotypes	Tumor Size, sm	
	<2	2–5
CD45–EpCAM+CK7–Snail–N-cadherin–Vimentin–	0.00 (0.00–3.32)	12.45 (1.66–17.43) *p* = 0.038
CD45–EpCAM+CK7–Snail–N-cadherin–Vimentin+	0.00 (0.00–0.00)	0.83 (0.83–2.49) *p* = 0.024
CD45–EpCAM–CK7+Snail–N-cadherin–Vimentin–	0.00 (0.00–0.00)	1.66 (0.83–9.96) *p* = 0.038

**Table 3 ijms-22-02504-t003:** The frequencies of 24 studied CTC phenotypes in groups of patients with and without lymphogenous metastasis.

	Lymph Node Metastasis	
	No	Yes
CD45–EpCAM–CK7+Snail–N-cadherin+Vimentin+	46.7 (14/30)	66.7 (6/9) *p* = 0.45
CD45–EpCAM–CK7+Snail+N-cadherin+Vimentin+	16.7 (5/30)	55.6 (5/9) *p* = 0.03
CD45–EpCAM+CK7+Snail–N-cadherin+Vimentin+	30.0 (9/30)	33.3 (3/9) *p* = 1.00
CD45–EpCAM+CK7+Snail+N-cadherin+Vimentin+	33.3 (10/30)	33.3 (3/9) *p* = 1.00
CD45–EpCAM–CK7+Snail–N-cadherin+Vimentin–	10.0 (3/30)	22.2 (2/9) *p* = 0.63
CD45–EpCAM+CK7+Snail–N-cadherin+Vimentin–	6.7 (2/30)	11.1 (1/9) *p* = 0.55
CD45–EpCAM–CK7+Snail+N-cadherin+Vimentin–	3.3 (1/30)	11.1 (1/9) *p* = 0.41
CD45–EpCAM+CK7+Snail+N-cadherin+Vimentin–	3.3 (1/30)	33.3 (3/9) *p* = 0.03
CD45–EpCAM+CK7–Snail–N-cadherin–Vimentin–	50.0 (15/30)	77.8 (7/9) *p* = 0.25
CD45–EpCAM+CK7–Snail+N-cadherin–Vimentin–	23.3 (7/30)	33.3 (3/9) *p* = 0.68
CD45–EpCAM+CK7–Snail–N-cadherin–Vimentin+	46.7 (14/30)	55.5 (5/9) *p* = 0.71
CD45–EpCAM+CK7–Snail+N-cadherin–Vimentin+	6.7 (2/30)	22.2 (2/9) *p* = 0.22
CD45–EpCAM–CK7+Snail–N-cadherin–Vimentin+	56.7 (17/30)	88.9 (8/9) *p* = 0.12
CD45–EpCAM–CK7+Snail–N-cadherin–Vimentin–	40.0 (12/30)	77.8 (7/9) *p* = 0.06
CD45–EpCAM–CK7+Snail+N-cadherin–Vimentin+	30.0 (9/30)	55.5 (5/9) *p* = 0.23
CD45–EpCAM–CK7+Snail+N-cadherin–Vimentin–	13.3 (4/30)	11.1 (1/9) *p* = 1.00
CD45–EpCAM+CK7+Snail–N-cadherin–Vimentin+	10.0 (3/30)	0.0 (0/9) *p* = 1.00
CD45–EpCAM+CK7+Snail–N-cadherin–Vimentin–	16.7 (5/30)	11.1 (1/9) *p* = 1.00
CD45–EpCAM+CK7+Snail+N-cadherin–Vimentin+	3.3 (1/30)	0.0 (0/9) *p* = 1.00
CD45–EpCAM+CK7+Snail+N-cadherin–Vimentin–	0.0 (0/30)	0.0 (0/9) *p*-no data
CD45–EpCAM+CK7–Snail–N-cadherin+Vimentin–	26.7 (8/30)	33.3 (3/9) *p* = 0.69
CD45–EpCAM+CK7–Snail+N-cadherin+Vimentin+	10.0 (3/30)	44.4 (4/9) *p* = 0.03
CD45–EpCAM+CK7–Snail+N-cadherin+Vimentin–	20.0 (6/30)	55.5 (5/9) *p* = 0.08
CD45–EpCAM+CK7–Snail–N-cadherin+Vimentin+	23.3 (7/20)	55.5 (5/9) *p* = 0.10

**Table 4 ijms-22-02504-t004:** The number of 24 studied CTC phenotypes in groups of patients with and without lymphogenous metastasis.

	Lymph Node Metastasis	
	No	Yes
CD45–EpCAM–CK7+Snail–N-cadherin+Vimentin+	1.24 (0.00–3.32)	1.24 (0.41–6.22) *p* = 0.86
CD45–EpCAM–CK7+Snail+N-cadherin+Vimentin+	0.00 (0.00–0.41)	0.83 (0.00–2.07) *p* = 0.19
CD45–EpCAM+CK7+Snail–N-cadherin+Vimentin+	0.00 (0.00–0.83)	0.00 (0.00–0.83) *p* = 0.67
CD45–EpCAM+CK7+Snail+N-cadherin+Vimentin+	0.41(0.00–1.24)	0.00 (0.00–2.90) *p* = 0.98
CD45–EpCAM–CK7+Snail–N-cadherin+Vimentin–	0.00 (0.00–0.00)	0.00 (0.00–0.41) *p* = 0.82
CD45–EpCAM+CK7+Snail–N-cadherin+Vimentin–	0.00 (0.00–0.00)	0.00 (0.00–0.00) *p* = 0.90
CD45–EpCAM–CK7+Snail+N-cadherin+Vimentin–	0.00 (0.00–0.00)	0.00 (0.00–0.00) *p* = 0.78
CD45–EpCAM+CK7+Snail+N-cadherin+Vimentin–	0.00 (0.00–0.00)	0.00 (0.00–0.83) *p* = 0.21
CD45–EpCAM+CK7–Snail–N-cadherin–Vimentin–	9.54 (0.41–17.43)	6.22 (2.49–11.20) *p* = 0.63
CD45–EpCAM+CK7–Snail+N-cadherin–Vimentin–	0.00 (0.00–0.83)	0.00 (0.00–4.15) *p* = 0.78
CD45–EpCAM+CK7–Snail–N-cadherin–Vimentin+	0.83 (0.00–2.90)	0.83 (0.00–1.24) *p* = 0.38
CD45–EpCAM+CK7–Snail+N-cadherin–Vimentin+	0.00 (0.00–0.00)	0.00 (0.00–0.41) *p* = 0.60
CD45–EpCAM–CK7+Snail–N-cadherin–Vimentin+	3.32 (0.83–7.07)	4.15 (1.66–6.22) *p* = 0.82
CD45–EpCAM–CK7+Snail–N-cadherin–Vimentin–	1.66 (0.00–7.05)	1.66 (0.83–2.90) *p* = 0.94
CD45–EpCAM–CK7+Snail+N-cadherin–Vimentin+	0.00 (0.00–1.24)	0.83 (0.00–1.66) *p* = 0.60
CD45–EpCAM–CK7+Snail+N-cadherin–Vimentin–	0.00 (0.00–0.00)	0.00 (0.00–0.00) *p* = 0.70
CD45–EpCAM+CK7+Snail–N-cadherin–Vimentin+	0.00 (0.00–0.00)	0.00 (0.00–0.00) *p* = 0.56
CD45–EpCAM+CK7+Snail–N-cadherin–Vimentin–	0.00 (0.00–0.41)	0.00 (0.00–0.00) *p* = 0.60
CD45–EpCAM+CK7+Snail+N-cadherin–Vimentin+	0.00 (0.00–0.00)	0.00 (0.00–0.00) *p* = 0.86
CD45–EpCAM+CK7+Snail+N-cadherin–Vimentin–	0.00 (0.00–0.00)	0.00 (0.00–0.00) *p*-no data
CD45–EpCAM+CK7–Snail–N-cadherin+Vimentin–	0.00 (0.00–1.24)	0.00 (0.00–0.83) *p* = 0.63
CD45–EpCAM+CK7–Snail+N-cadherin+Vimentin+	0.00 (0.00–0.00)	0.41 (0.00–1.66) *p* = 0.19
CD45–EpCAM+CK7–Snail+N-cadherin+Vimentin–	0.00 (0.00–0.83)	0.83 (0.00–1.24) *p* = 0.21
CD45–EpCAM+CK7–Snail–N-cadherin+Vimentin+	0.00 (0.00–0.83)	1.24 (0.00–1.66) *p* = 0.18

**Table 5 ijms-22-02504-t005:** The number of 24 studied CTC phenotypes in groups of patients with and without lymphogenous metastasis.

Parameter		Frequency, % (n)
Age	<35	5.1 (2/39)
	35–50	30.8 (12/39)
	>50	64.1 (25/39)
Menstrual function	Premenopausal	33.3 (13/39)
	Postmenopausal	66.7 (26/39)
Tumour size	<20 mm	23.1 (9/39)
	20–50 mm	71.8 (28/39)
	No data	5.1 (2/39)
Stage	I	28.2 (11/39)
	IIA	53.8 (21/39)
	IIB	18.0 (7/39)
Molecular subtype	Luminal A	33.3 (13/39)
	Luminal B	59.0 (23/39)
	Triple negative	5.1 (2/39)
	Her2-positive	2.6 (1/39)
Tumour grade	1	23.1 (9/39)
	2	69.2 (27/39)
	3	7.7 (3/39)
Estrogen receptor	positive	92.3 (36/39)
	negative	7.7 (3/39)
Progesteron receptor	positive	66.7 (26/39)
	negative	33.3 (13/39)
HER2/neu	positive	28.2 (11/39)
	negative	71.8 (28/39)
Ki67 expression	<20%	46.2 (18/39)
	>20%	53.8 (21/39)
Lymph node metastasis	Yes	23.1 (9/39)
	No	76.9 (30/39)
Distant metastasis	Yes	0.0 (0/39)
	No	100.0 (39/39)
Intraoperative radiation therapy	No	20.5 (8/39)
	Yes	69.2 (27/39)
	No data	10.3 (4/39)

## Data Availability

The data presented in this study are available on request from the corresponding author. The data are not publicly available due to data is being prepared for intellectual property approval.

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
