# Peer review of "Heterogeneous Manifestations of Epithelial–Mesenchymal Plasticity of Circulating Tumor Cells in Breast Cancer Patients"

_ijms, 2021, doi:10.3390/ijms22052504_

Round 1
Reviewer 1 Report
Authors should describe the most recent progress of breast cancer CTCs in introduction, and discuss how this work is relevant to others. One of the recent publications is "Liu et al., 2019 Sci Adv 5:eaav4275".
Reviewer 2 Report
General considerations
This is a prospective and descriptive clinical study of the detection of CTCs in a series of patients diagnosed with newly diagnosed breast cancer. The aim of the study is to characterize CTCs from the point of view of epithelial-mesenchymal phenotype. The study also aims to establish the clinical significance of either the number or the phenotype of the CTCs in breast cancer at the time of diagnosis.
Major aspects
Material and Methods
The authors want to know the clinical significance of either the number or the phenotype of the CTCs through clinical correlations. However, it seems there is not enough number of cases to get this objective. Given these difficulties, how it is possible to be sure about the true value of the statistical analysis?
The mechanisms of metastasis through the blood and lymphatics are different. Then, the objective to compare the number and type of CTCs in the blood with the development of lymph node metastasis has not a previous rationale. The authors should clear up which is the argumentation of this comparison.
Results
The authors state that the results of the study indicate that most of the CTCs exhibit epithelial-mesenchymal phenotype. However, in a previous study (Markiewicz, A et al. Spectrum of Epithelial-Mesenchymal Transition Phenotypes in Circulating Tumor Cells from Early Breast Cancer Patients. Cancers 2019, 11, 59) with a larger sample this phenotype was not dominant. What explanation do the authors have for this discrepancy? (Line number 504).
Conclusions
The Conclusions are confusing, long, and raise questions that should not be written in this section of the article. All conclusions must be rewritten.
The authors state that “significant interpersonal heterogeneity gives hope for the identification of CTC phenotypes that are most consistent with the phenotype of the "seeds"”, is a general unsubstantiated statement that should be deleted. (Line number 27, abstract).
The authors´ claim that “the combinations of different CTC phenotypes were not the same in different patients” does not provide knowledge and should be eliminated. (Line number 26, abstract).
Minor aspects
Title
In general, acronyms cannot appear in the title. Then, it is not acceptable the acronym CTCs in the title of the article. It must be written as Circulating Tumor Cells. (Line number 3).
Material & Methods
It should be specified that type of breast carcinomas, ductal or lobular, authors are studying on. (Line number 429).
Table
What is the meaning of IORT? This acronym must be previously defined (Line number 532).
Round 2
Reviewer 2 Report
The authors have responded to the observations made in a consistent manner. Consequently, the authors have rectified the manuscript in an acceptable way.
The "Conclusion" section does not conform to the length of a scientific article, as indicated in the previous review. The authors unnecessarily repeat aspects related to results and discussion. The conclusions should be very concise and highlight exclusively what the article contributes to what is already known (one or two paragraphs maximum).
Round 3
Reviewer 2 Report
The manuscript provides a new version with an inadequate presentation of the Conclusions. The authors have sent the original text partially crossed out. This is not the correct way to submit a manuscript. Authors must send the final version in which only the selected text appears in Conclusions.
Author Response
Dear Reviewer! Could you specify what exactly should be corrected in the Conclusion in your opinion? We have removed phrases repeated results and discussions and made the conclusion as brief and clear as possible. The latest version of Conclusion looks this way:
"Conclusions
The results of this work indicate that the CTCs in breast cancer are characterized by pronounced heterogeneity in the manifestations of the EMT. Heterogeneity can be both intra - and interpersonal. Taking into account the presence or absence of epithelial and mesenchymal markers used in our study, most of the detected CTC phenotypes can be attributed to hybrid ones. The wide range of different EMT phenotypes of CTC suggests that the degree of EMT is not an obstacle to intravasation. However, we have shown that the number of CTCs belonging to different populations is largely determined by their phenotype and is not proportional to the number of similar populations in the primary tumor, which may be due to the different ability to intravasate and survive in the blood. The association between some СTC phenotypes and tumor size, proliferative activity, and lymphogenic metastasis suggests that, as the duration of prospective follow-up increases, it will be possible to identify СTC phenotypes associated with hematogenous metastasis from the detected ones."
We would like to make our publication meets all the requirements of the journal and would be grateful for your future explanations.
Thank you in advance,
Evgeniya Grigoryeva
Round 4
Reviewer 2 Report
As it was explained, in the last version of the manuscript that I had received, the text of Conclusions was written in a preliminary way with unacceptable crossed out sentences. In this new version of the manuscript Conclusions are brief, clear and sound. I have not got new criticisms of the manuscript.